# Susceptibility Patterns in *Staphylococcus* and *Klebsiella* Causing Nosocomial Infections upon Treatment with *E*-Anethole-Rich Essential Oil from *Clausena anisata*

François Nguimatsia [1], Evariste Josué Momo [2,3], Paul Keilah Lunga [4], Virginia Lum Tamanji [1], Boniface Pone Kamdem [4,*] and Pierre Michel Jazet Dongmo [3]

[1] Laboratory of Pharmacognosy, Faculty of Pharmacy, University of the Mountain, Bangangté P.O. Box 208, Cameroon
[2] Higher Teacher Training College, Bertoua (HTTC), University of Bertoua, Bertoua P.O. Box 652, Cameroon; momoevaj@yahoo.fr
[3] Laboratory of Biochemistry, Faculty of Science, University of Douala, Douala P.O. Box 24157, Cameroon
[4] Antimicrobial and Biocontrol Agents Unit (AmBcAU), Laboratory for Phytobiochemistry and Medicinal Plants Studies, Department of Biochemistry, Faculty of Science, University of Yaounde I, Yaounde P.O. Box 812, Cameroon
* Correspondence: ponekamdemboniface@gmail.com; Tel.: +237-680-98-76-69

**Abstract:** High rates of resistance to antibiotics are associated with healthcare-related infections, thus demonstrating the urgent need for effective antimicrobials against these maladies. The present study aims to determine the chemical composition of essential oil (EO) from *Clausena anisata* leaves and evaluate their antibacterial activity against selected nosocomial bacteria. To this end, one kilogram (1 kg) of fresh leaves of *C. anisata* was washed and boiled with 500 mL of distilled water for 2–4 h using a Clevenger apparatus. The oil was then collected in an Erlenmeyer, dried using anhydrous sodium sulfate, bottled in a tinted glass bottle and refrigerated at 4 °C before analysis. Next, the as-prepared oil was analyzed using gas chromatography-mass spectrometry (GC-MS). The essential oil was further tested against a panel of selected nosocomial bacteria, including *Staphylococcus* and *Klebsiella* species, among others, by microdilution using a resazurin assay to determine the minimum inhibitory and minimum bactericidal concentrations (MICs and MBCs, respectively). As a result, 0.77% of EO was extracted from fresh leaves of *C. anisata*. The GC-MS analysis revealed that the as-prepared essential oil contained E-anethole (70.77%), methyl isoeugenol (13.85%), estragole (4.10%), γ-terpinene (3.33%), myrcene (2.82%) and sabinene (0.77%), with E-anethole being the major constituent. Twenty-two compounds were identified in the EO of *C. anisata* leaves through gas chromatography. Upon antibacterial testing against selected nosocomial pathogens, the *E*-anethole-rich essential oil exhibited MIC and MBC values ranging from 3.91 to 125 μg/mL and 7.81 to 125 μg/mL, respectively, indicative of a bactericidal orientation of the plant's essential oil (MIC/MBC ratio < 4). This novel contribution highlights the scientific validation of the use of *C. anisata* leaves in the traditional treatment of various infectious diseases. However, toxicity and pharmacokinetic studies, mechanistic bases of the antibacterial action, and in vivo antibacterial experiments of the *E*-anethole-rich EO of *C. anisata* should be investigated to successfully use this plant in the treatment of infectious diseases.

**Keywords:** nosocomial infections; antibacterial activity; drug resistance; *Clausena anisata*; GC-MS analysis; essential oil

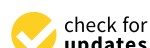



## 1. Introduction

Infectious diseases are a group of illnesses caused by specific infectious agents—also called pathogens—or their toxic products through an intermediate host, vector or inanimate environment [1]. Among these infectious agents are nosocomial pathogens, which are acquired during the process of receiving health care and are not present at the time of

admission [2]. During hospitalization, patients are exposed to pathogens through different sources, including the environment, healthcare staff and other infected patients [2,3]. Nosocomial infections occur worldwide both in developed and developing countries and are the cause of high mortality and morbidity. These maladies account for 7% in developed and 10% in developing countries of the total [2]. According to WHO estimates, approximately 15% of all hospitalized patients suffer from these infections [4]. These diseases are responsible for over 4–56% of all deaths caused in neonates, with an incidence rate of 75% in Southeast Asia and Sub-Saharan Africa [2]. The incidence of nosocomial infections is high enough in high-income countries, specifically, between 3.5% and 12%, whereas the same varies between 5.7% and 19.1% in middle- and low-income countries [2]. The most frequent types of nosocomial infections include central line-associated bloodstream infections, catheter-associated urinary tract infections, surgical site infections and ventilator-associated pneumonia [5,6]. Among these, surgical site infections and ventilator-associated pneumonia are mainly caused by *Staphylococcus aureus* and *Klebsiella pneumonia*, respectively, resulting in prolonged hospitalization and risk of death for admitted patients [2,7]. Transmission of such germs happens in hospitals, nursing homes and other places with lots of sick people or through the touch of a cut during handshaking or by contact with medical devices [2,8]. The high resistance of *Klebsiella* species (Enterobacteriaceae) to carbapenems and *Staphylococcus* species to methicillin make these pathogens highly virulent, especially when they travel from the gastrointestinal tract (opportunistic pathogen, *Klebsiella*) to other parts of the body [9,10] or when they are acquired through direct contact with open wounds and contaminated hands (*Staphylococcus aureus*; [2]). The Gram-negative bacteria *Klebsiella*, Enterobacter and *Serratia* are closely related to normal intestinal flora that rarely cause disease in normal hosts. *Serratia* species are Gram-negative bacilli of the Enterobacterales order, although they are not a common component of healthy human fecal flora [11]. However, *Serratia* species may harbor multidrug resistance mechanisms that can complicate treatment resolutions [11,12]. Current treatments for nosocomial diseases include carbapenems (meropenem) and beta lactams (tazobactam and avibactam) as well as combination therapies, such as meropenem–vaborbactam, ceftolozane–tazobactam, imipenem–cilastatin/relebactam, dmeropenem–vaborbactam, ceftazidime–avibactam and aztreonam–avibactam [13,14]. However, the misuse and overuse of these antimicrobials are the main drivers of the development of drug-resistant pathogens [13,15]. Moreover, a number of adverse effects [(carbapenems: injection site reactions, diarrhea, nausea, coagulation abnormalities, nephrotoxicity or hepatotoxicity, vomiting and skin rashes [16,17]; beta lactams: neurotoxicity and nephrotoxicity as well as hematological adverse events and drug-induced liver injury [18]; cephalosporins: cefepime-induced neurotoxicity [19]] have been identified for these therapies. Thus, there is an urgent need to search for safe and effective antimicrobials against nosocomial infections caused by bacteria.

Medicinal plants, which have been the most valuable source of molecules with therapeutic potential, are still an important pool for the identification of new and safe drugs, including antimicrobials [20]. It is even expected that plant extracts showing target sites other than those used by antibiotics will be active against drug-resistant pathogens [21]. Various parts of *Clausena anisata* have been reported as effective treatments against parasitic infections, especially flatworm infections, such as taeniasis and schistosomiasis, and eye diseases; influenza and other respiratory illnesses [22]; hypertension; heart diseases; and abdominal cramps. Moreover, other reports indicated *C. anisata* as an effective plant against liver and gastrointestinal problems, fevers and pyrexia, malaria, boils, rheumatism and other inflammatory diseases, toothaches, body pains, headaches, swollen gums, impotence, convulsions, sterility, and several mental disorders [23]. The leaves of *C. anisata* are used to treat hypertension in South Africa, and fresh leaves are burned to repel mosquitoes in the Philippines [24,25]. For generations, different parts of *Clausena* plants, including root, bark, stem and leaves as well as essential oil (EO), which are rich in alkaloids, flavonoids, monoterpenes and triterpenoids, have traditionally been used to treat a number of diseases while being a source of drug for modern-day microbial diseases [26]. In Senthilkumar

and Venkatesalu's [23] paper, an analysis of *C. anisata* essential oil using gas chromatography (GC) and GC coupled with mass spectrometry (GC-MS) revealed sabinene (33.0%), germacrene-D (17.0%), Z-β-ocimene (6.0%), germacrene-B (5.5%), (E)-β-ocimene (4.9%) and terpinen-4-ol (4.7%) as the major constituents. Upon GC-MS analysis, the leaf essential oil of *C. anisata*, a species growing in North-Central Nigeria, was found to be rich in anethole (31.1%), followed by trans-β-ocimene (20.0%), β-elemene (10.5%), estragole (6.9%), α-pinene (6.7%) and γ-cadinene (5.4%) [24,27]. In 2014, Njonkep [28] used the GC-MS technique to identify trans-anethole (69.3%), methyl iso-eugenol (13.2%), ɣ-terpinene (4.4%), myrcene (3.8%), estragole (2.3%), β-phellandrene (1.2%), β-caryophyllene (0.8%), germacrene-D (0.7%) and methyleugenol (0.5%) as the major constituents of the EO of *C. anisata* leaves from Cameroon [28]. Furthermore, the essential oil obtained from various parts (leaves, stems, bark and roots, etc.) of plants from the genus *Clausena* were found to contain monoterpenes and triterpenoids [29]. According to these studies, there is no denying that a number of terpenoids have been identified thus far in the EO of *C. anisata*; however, classical or traditional methods, such as hydro-distillation (water and steam distillations), were used for the extraction of the EO of *C. anisata*. Although classical methods of extraction offer several benefits, including cheap processing costs, ease of use, excellent performance, complete extract recovery and reduced time and solvent consumption [30–32], robust techniques (ultrasound-assisted and microwave-assisted extractions, pressurized liquid and supercritical fluid extractions) of essential oil extraction should be anticipated since they display better efficiency and shorter extraction times [33,34]. Nonetheless, the high cost and unavailability of these modern techniques, especially in less affordable laboratory settings, are noteworthy. On the other hand, modern pharmacological studies revealed the antimicrobial activity of the EO of *C. anisata* on *Salmonella typhi* and *Pseudomonas aeruginosa* [23]. In a paper published by Agyepong et al. [35], ethanol and methanol extracts exhibited mild anti-*Bacillus subtilis* activity, with MIC values of 500 and 600 μg/mL, respectively. In addition, Lawal et al. [36] described a moderate antibacterial activity of the acetone extract of *C. anisata* leaves against *Streptococcus pyogenes* (MIC: 100 μg/mL), *Staphylococcus aureus* (ATCC 6538) (MIC: 100 μg/mL) and *Bacillus cereus* (ATCC 10702) (MIC: 500 μg/mL), whereas De Canha et al. [37] reported the inhibitory effects of *C. anisata* ethanol leaf extract on *Propionibacterium acnes* (MIC value: 62.5 μg/mL). Aqueous, ethanol and chloroform extracts (dose: 39–117 mg/kg) from *C. anisata* leaves demonstrated in vivo antiplasmodial and analgesic activities in mice models [38]. More recently, two compounds (heptaphylline and imperatorin), which were isolated from *C. anisata* bark, revealed antiplasmodial activity against trophozoite ($IC_{50}$ values: 1.57 and 2.23 μM), schizont ($IC_{50}$ values: 8.97 and 6.71 μM) and gametocyte ($IC_{50}$ values: 26.92 and 20.87 μM) stages of *Plasmodium falciparum* [39]. However, there is not yet any scientific evidence of the effect of the EO of *C. anisata* against nosocomial infections caused by *Staphylococcus* and *Klebsiella* species. Thus, the present study demonstrates the antibacterial activity of *C. anisata* EO against selected nosocomial pathogens, including clinical strains of *Staphylococcus*, *Klebsiella* and *Bacillus* species, among others.

## 2. Results and Discussion

### 2.1. Results

2.1.1. Extraction of the Essential Oil

The physical characteristics observed on the EO included the color, consistency and odor. The obtained EO was yellowish-brown and oily with an aniseed aroma. From 4600 g of fresh leaves of *C. anisata*, 23.46 g of essential oil was obtained to afford a yield of 0.51%.

2.1.2. Chemical Composition

Upon analysis of the EO by using gas chromatography, a chromatogram that was generated showed compound peaks at different retention times (Figure S1, Supplementary Material). The values obtained for retention times were used to generate Kovats indices through calculations using the formula highlighted in the material and methods' section.

The Kovats index obtained for each peak facilitated the identification of the chemical compounds of the EO from *C. anisata* fresh leaves.

Upon GC-MS analysis of the EO of *C. anisata* leaves, it was found to contain E-anethole (70.77%), methyl isoeugenol (13.85%), estragole (4.10%), γ-terpinene (3.33%), myrcene (2.82%) and sabinene (0.77%) (Table 1).

**Table 1.** Chemical composition of the essential oil of *Clausena anisata*.

| Serial N° | RT | IK | Identified Compound | Percentage (%) |
|---|---|---|---|---|
| 1 | 10.547 | 933 | α-Pinene | 0.26 |
| 2 | 12.062 | 971 | Sabinene | 0.77 |
| 3 | 12.270 | 976 | β-Pinene | 0.26 |
| 4 | 12.640 | 986 | Myrcene | 2.82 |
| 5 | 14.195 | 1021 | p-Cymene | 0.52 |
| 6 | 14.407 | 1025 | Limonene | 0.77 |
| 7 | 15.118 | 1041 | E-β-Ocimene | 0.26 |
| 8 | 15.795 | 1055 | γ-Terpinene | 3.33 |
| 9 | 17.153 | 1084 | Terpinolene | 0.25 |
| 10 | 21.515 | 1171 | α-Terpineol | Nd |
| 11 | 22.562 | 1191 | Estragole | 4.10 |
| 12 | 25.115 | 1242 | para Anisaldehyde | 0.25 |
| 13 | 25.497 | 1250 | Z-Anethole | 0.25 |
| 14 | 27.515 | 1291 | E-Anethole | 0.77 |
| 15 | 28.195 | 1305 | Cinamyl alcool | Nd |
| 16 | 32.175 | 1387 | Cinamylacetate | 0.52 |
| 17 | 33.395 | 1413 | Methyleugenol | 0.52 |
| 18 | 34.552 | 1438 | β-Caryophyllene | Nd |
| 19 | 34.922 | 1447 | Isoeugenol | 0.25 |
| 20 | 36.123 | 1473 | γ-Gurjunene | 0.50 |
| 21 | 36.780 | 1487 | Methyl iso eugenol | 13.85 |

IK: Index of Kovats; Nd: Not determined; RT: Retention time.

### 2.1.3. Antibacterial Activity

The evaluation of the antibacterial activity of the EO from *C. anisata* fresh leaves on eight bacterial strains that are incriminated in nosocomial infections led to the determination of the minimum inhibitory concentrations (MICs) and minimum bactericidal concentrations (MBCs), as indicated in Table 2. The incubation of the bacterial strains with the EO afforded MIC and MBC values ranging from 3.91 to 125 μg/mL and from 7.81 to 125 μg/mL, respectively. Except for *Pseudomona aeruginosa* and *Salmonella typhimurium*, the EO of *C. anisata* leaves was found to be bactericidal for all the nosocomial pathogens tested as the calculated MBC/MIC ratios were found to be less than 4 (Table 2). It is well known that natural products (plant extracts and compounds, essential oils, etc.) reveal a bactericidal orientation when the MBC/MIC ratio is less than 4 [40–43]. Consequently, the EO from *C. anisata* leaves revealed bactericidal orientation against *Klebsiella* and *Staphylococcus* species, the pathogens that are responsible for most of the nosocomial diseases. Against the *Bacillus* species, the EO revealed a common value for MIC and MBC (31.25 μg/mL). Moreover, *Salmonella typhimurium* and *Escherichia coli* were inhibited with a common MIC value of 31.25 μg/mL, whereas the MBC values obtained against these pathogens were respectively 125 and 62.5 μg/mL.

**Table 2.** Antibacterial activity of the essential oil from *C. anisata* leaves.

| Bacterial Strains | MIC (μg/mL) | MBC (μg/mL) | MIC/MBC |
|---|---|---|---|
| *Bacillus* spp. | 31.25 | 31.25 | 1 |
| *Staphylococcus aureus* | 15.63 | 15.63 | 1 |
| *Klebsiella pneumoniae* | 3.91 | 15.63 | 4 |
| *Pseudomonas aeroginosa* | 62.5 | 62.5 | 1 |
| *Staphylococcus epidermidis* | 3.91 | 7.81 | 2 |
| *Serratia* spp. | 3.91 | / | / |
| *Escherichia coli* | 31.25 | 62.5 | 2 |
| *Salmonella typhimurium* | 31.25 | 125 | 4 |

MIC: Minimum inhibitory concentration; MBC: Minimum bactericidal concentration.

*2.2. Discussion*

The present study aims to evaluate the antibacterial activity of essential oil from fresh leaves of *C anisata* against selected nosocomial pathogens. The EO was obtained using a Clevenger apparatus as a yellowish-brown oil, with a smell similar to that of aniseed [44]. Almost similar physical characteristics were obtained by Yaouba et al. [45] while working on the antifungal activity of EO from *C. anisata* leaves. The yield of extraction of the EO was 0.77%, a value that was found to be superior to that (0.55%) obtained by Okokon [38] in 2012 while working on the EO of *C. anisata* leaves collected from Nigeria [38], thus highlighting potential differences in the chemical compositions of EO from *C. anisata* leaves that are collected in various areas. Upon analysis of the essential oil from *C. anisata* leaves through the GC/MS technique, its chemical constituents included E-anethole (70.77%), methyl isoeugenol (13.85%), estragole (4.10%), γ-terpinene (3.33%), myrcene (2.82%) and sabinene (0.77%). This chemical composition of the oil is quite different from that [trans-anethole (69.3%), methyl isoeugenol (13.2%), ɣ- terpinene (4.4%), myrcene (3.8%), estragole (2.3%), β-phellandrene (1.2%), β-caryophyllene (0.8%), germacrene-D (0.7%) and methyl eugenol (0.5%)] obtained by Njonkep in 2014 [28] for *C. anisata* leaves collected in the same area (Bafou, Dschang, West Region of Cameroon). These results suggest the impact of seasons and climatic conditions on the chemical composition of essential oil extracted from plants [28]. The effect of the time of plant collection and extraction, as well as extraction conditions, on the yield of extraction is noteworthy. In fact, the chemical composition of EO from *C. anisata* leaves differs according to the place of plant harvest [Mount Bamboutos (major constituent: 93.1% of myrtenyl acetate; absence of estragole) [46], Ngaoundere (*C. anisata* leaf essential oil deprived of estragole) [45]]. Thus, it can be speculated that the chemical compounds of any plant's essential oil vary greatly depending on the geographical region, the age of the plant and local climatic, seasonal and experimental conditions [47]. Genetic differences are also responsible for the changes in chemical compounds, thereby altering the studied biological activities [48]. These differences in the chemical composition of EO might significantly impact the antimicrobial efficacy. In this study, the incubation of the bacterial strains with *C. anisata* leaf essential oil afforded MIC and MBC values ranging from 3.91 to 125 μg/mL and from 7.81 to 125 μg/mL, respectively. These results suggested a bactericidal orientation of the EO as the MIC/MBC ratio was found to be less than 4. A similar trend was reported by several authors [49,50] after the determination of MIC and MBC correlations following the incubation of selected bacteria with extracts and/or essential oil from plants. Other authors reached similar conclusions while working on the influence of natural and synthetic compounds vis-à-vis the kinetics of bacterial mortality as a function of time (time-kill kinetics) [51–55]. The observed antibacterial activity might be attributed to the presence of a number of monoterpenes (E-anethole, methyl isoeugenol, myrcene sabinene and γ-terpinene) in the plant, which were mainly dominated by E-anethole (70.77%). In fact, these compounds are well known for their antibacterial activity, although their mechanism of action is poorly understood. Indeed,

in a study by Senatore et al. [56], a trans-anethole-rich oil from another plant species, i.e., *Foeniculum vulgare* leaves, exhibited antibacterial activity against a series of bacterial strains, including *Bacillus subtilis*, *Staphylococcus aureus*, *Staphylococcus epidermidis*, *Streptococcus faecalis*, *Escherichia coli* and *Klebsiella pneumoniae*, etc., attesting to the potential of anethole to significantly inhibit the growth of several bacteria. Other studies [57–59] have also revealed the antibacterial activity of trans-anethole against numerous bacterial strains. Previous studies have also demonstrated the inhibitory effect of *E*-anethole against some fungal strains, such as *Saccharomyces cerevisiae* [47]; however, the antibacterial activity of E-anethole has not yet been reported. Nevertheless, it is speculated that the antibacterial mechanisms of action of these lipophilic compounds involve bacterial membrane disruption [60–63]. The antibacterial activity of monoterpene-rich essential oil has been attributed to the inhibition and eradication of biofilm formation [58,64] and increases in membrane permeability, which allow higher amounts of test samples to enter pathogenic cells and consequentially destroy them [65]. In addition, Li et al. [66] demonstrated the antibacterial activity (MIC values: micromolar range) of a number of monoterpenes isolated from *Illicium simonsii* stems and leaves. Li's group attributed the observed antibacterial activity of *Illicium simonsii* monoterpenes to the disruption of the bacterial membrane permeability, as revealed by 4′,6-diamidino-2-phenylindole (DAPI) and propidium iodide (PI) assays [66]. A number of previous reports [67–69] have also pointed out efflux pump as one of the mechanisms by which monoterpene-rich essential oils exert antibacterial activity.

To our knowledge, this is the first report on the inhibitory potential of an E-anethole-rich essential oil from *C. anisata* leaves against *Staphylococcus* and *Klebsiella* causing nosocomial infections. The present study validates the traditional use of *C. anisata* leaves in the treatment of various infectious diseases.

### 3. Material and Methods

*3.1. Material*

3.1.1. Plant Material

*Clausena anisata* leaves (Figure 1) were harvested at Bafou, Dschang, Menoua division (West Region of Cameroon). The plant was later identified at the National Herbarium of Cameroon (NHC) in Yaounde, where a voucher specimen was deposited under number 2711/SRFK.

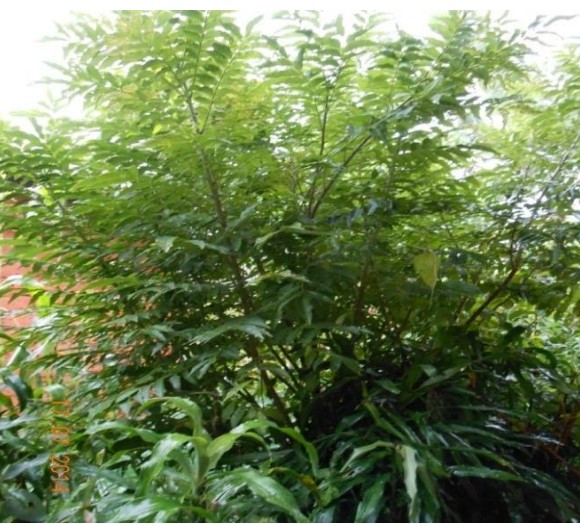

**Figure 1.** Photograph of *Clausena anisata* growing on a fence in Bafou, Dschang, West Region, Cameroon (picture by V.L.T.).

3.1.2. Bacterial Strains

The bacterial strains used in this study included nosocomial pathogens, such as *Bacillus* spp. (3C4 UR Mcc), *Staphylococcus aureus* (00166 6/14), *Klebsiella pneumoniae* (Rea Ro2Mcc),

*Serratia* spp. (2C1 UR CB Mcc), *Pseudomonas aeruginosa* (2′), *Staphylococcus epidermidis* (3), *Escherichia coli* (2.5922) and *Salmonella typhimurium* (O,6,7 HCN).

### 3.1.3. Material for Bacterial Cell Culture

In this study, Mueller Hinton Agar was used for the development of the bacterial strains, whereas Mueller Hinton Broth was employed for the determination of the minimum inhibitory concentration (MIC) and minimum bactericidal concentration (MBC). These media were obtained from Liofilchem® S.r.L (Scozia, Zona Industriale 64026, Roseto degli Abruzzi, Italy). Other reagents included McFarland standard 0.5, sterile distilled water, physiological water (normal saline), tween 80 and anhydrous sodium sulfate (Sigma-Aldrich, Darmstadt, Germany).

### *3.2. Methods*

### 3.2.1. Extraction of the Essential Oil

The essential oil was extracted from fresh leaves of *C. anisata* using a Clevenger-type apparatus. Briefly, the collected plant material was washed and then chopped. Next, the plant material was introduced to a round bottom flask, with 1 kg for every 500 mL of water. The mixture was then brought to a boil for a period of 4 to 5 h. During this process, the vapor underwent condensation and was divided into 2 phases, with the superior phase consisting of the EO, which was collected. The water contained in the essential oil was then dried using anhydrous sodium sulfate. The oil was further weighed, and the yield was calculated and bottled in a tinted glass 60 mL bottle and refrigerated at 4 °C. Then, the yield of the EO was expressed as a percentage that was calculated using the following formula:

$$Y = (Me/Mp) \times 100$$

where

Y = yield of essential oil in percentage
Me = mass of essential oil in grams
Mp = mass of plant biomass in grams

### 3.2.2. GC-MS Analysis of Essential Oil of *C. anisata*

The essential oil was analyzed by gas chromatography (GC) on a Varian CP-3380 GC with a flame ionization detector fitted with a fused silica capillary column (30 m × 0.25 mm coated with DB5, film thickness 0.25 μm), with a temperature program of 50–200 °C at 5 °C/min, injector temperature of 200 °C and detector temperature of 200 °C with $N_2$ as the carrier gas (flow rate: 1 mL/min); then, gas chromatography coupled with mass spectrometry (GC-MS) was conducted using a Hewlett-Packard apparatus equipped with an HP1 fused silica column (30 m × 0.25 mm, film thickness 0.25 μm), interfaced with a quadrupole detector (GC-quadrupole MS system, model 5970). For GC-MS, the column temperature was programmed from 70° to 200 °C at 10 °C/min, whereas the injector temperature was set at 200 °C. Helium was used as the carrier gas at a flow rate of 0.6 mL/min, and the mass spectrometer was operated at 70 eV [70]. The linear retention indices of the compounds were relatively determined by the retention times of a series of n-alkanes, and the percentage compositions (Table 1) were obtained from electronic integration measurements, without taking into account the relative response factors [27,70]. After analysis by GC/GC-MS, the identification of different constituents of the oil was confirmed by a comparison of retention times and mass spectra with known values reported in the literature [70,71]. For each compound identified, the retention index (Kovats retention index) was determined according to the following formula:

$$IK = 100 \left[ n + \frac{Tr(x) - Tr(Cn)}{Tr(Cn + 1) - Tr(Cn)} \right]$$

IK = Kovats retention index

Tr (Cn) = retention time of alkane at n atoms of carbons
Tr (Cn + 1) = retention time of alkane at (n + 1) atoms of carbons
Tr (x) = retention time for compound x

### 3.2.3. Antibacterial Activity

a. Preparation of microbial inocula

To prepare the microbial inocula, single colonies from a 24 h old bacterial culture on MHA were separately isolated using a sterile loop and aseptically introduced into corresponding test tubes containing 10 mL of sterile normal saline (NaCl 0.9%). The turbidity of each test tube was adjusted to a 0.5 McFarland standard equivalent of $1.5 \times 10^8$ CFU/mL for bacteria. Microbial suspensions were diluted using Mueller Hinton Broth (MHB) to give final test concentrations of $1 \times 10^6$ CFU/mL.

b. Determination of minimum inhibitory concentrations (MICs) and minimum bactericidal concentrations (MBCs)

The various antimicrobial parameters (MIC and MBC) of the samples were evaluated using the broth microdilution method as previously described in protocol M07-A9 of the Clinical and Laboratory Standard Institute [72], with some modifications.

b.1. Determination of minimum inhibitory concentrations

Herein, the tests were performed in duplicate in sterile 96-well microtiter plates. Initially, 196 μL of Mueller Hinton Broth (MHB) was dispensed into the wells of the first line and 100 μL into the remaining wells. Four microliters (4 μL) of each test sample was added to the first wells and serial two-fold dilutions were made up to the eleventh wells. Next, 100 μL of $1 \times 10^6$ CFU/mL of bacterial suspensions were added to the wells to obtain a final volume of 200 μL. The medium and the corresponding microbial suspensions constituted the negative control, while the sterility control contained a culture medium only. Ciprofloxacin was used as the positive control. The final concentrations ranged from 1000 to 0.048 μg/mL for the test samples and from 3.906 to 0.0038 μg/mL for ciprofloxacin. Afterward, the plates were covered and incubated at 37 °C for 24 h. Following incubation, 20 μL of freshly prepared resazurin (0.15 mg/mL) was added to all wells and the plates were further incubated in the same conditions for 30 min. After this incubation period, the lowest concentration, which showed no visible color change from blue to pink, implied an absence of microbial growth and was considered as the minimum inhibitory concentration.

b.2. Determination of the minimum bactericidal concentrations

To determine the microbiostatic (bacteriostatic) or microbicidal (bactericidal) nature of the test samples, their MBCs were evaluated through the liquid subculture of the preparations withdrawn from the microplates initially used for the determination of MICs. After incubation of the microplates used for the determination of MICs, 25 μL aliquots of inhibitory wells were withdrawn and transferred into corresponding wells of sterile microplates containing 175 μL of sample-free broth per well. Hence, the quantities of samples in the various wells were diluted 8 times to eliminate their inhibitory effect. The microplates were then covered and incubated at 37 °C for 24 h. Following this incubation time, the minimum bactericidal concentration of each sample was determined using resazurin (0.15 mg/mL) (Sigma-Aldrich, Darmstadt, Germany) as previously described by the Clinical Laboratory Standard Institute [72]. From the MBC and MIC values, the ratio of MBC/MIC was calculated to depict the antibacterial orientation of the essential oil prepared from *C. anisata*.

### 3.3. Statistical Analysis

To determine the MIC and MBC values, concentration–response curves were designed using the Prism software package 5.00 for Windows, GraphPad Software, San Diego, CA, USA, www.graphpad.com (accessed on 1 May 2023) (GraphPad, San Diego, CA, USA), and data were reported as mean and standard deviation (SD) values obtained from a minimum of three determinations. The non-linear best fit was plotted with the SD and 95% confidence interval.

## 4. Conclusions

The treatment of nosocomial infections has been hampered by the emergence of drug resistance; thus, there is an urgent need to search for effective treatments against healthcare-associated pathogens. In this study, the antibacterial activity of an E-anethole-rich essential oil from *Clausena anisata* leaves was evaluated against *Staphylococcus* and *Klebsiella*, which cause healthcare-associated infections. E-anethole-rich essential oil from *C. anisata* leaves, which was obtained by steam distillation using a Clevenger apparatus, was further analyzed by GC-MS. As a result, the EO from *C. anisata* leaves (yield of extraction: 0.77%) was found to be majorly dominated by the bicyclic monoterpenoid E-anethole. The as-prepared essential oil inhibited the growth of *Staphylococcus* and *Klebsiella* species, with MIC and MBC values ranging from 3.91 to 125 μg/mL and from 7.81 to 125 μg/mL, respectively, suggesting a bactericidal orientation of this essential oil. This novel contribution highlights the scientific validation of the use of *C. anisata* leaves in the traditional treatment of numerous infectious diseases. However, toxicity and pharmacokinetic studies, mechanistic bases of the antibacterial action, and in vivo antibacterial experiments of the EO of *C. anisata* are warranted.

**Supplementary Materials:** The following supporting information can be downloaded at https://www.mdpi.com/article/10.3390/ddc3010014/s1: Figure S1: Chromatogram of the essential oil of *Clausena anisata*.

**Author Contributions:** Conceptualization, F.N., B.P.K. and P.M.J.D.; methodology, V.L.T. and F.N.; software, E.J.M., F.N., P.K.L. and V.L.T.; validation, E.J.M., P.K.L. and B.P.K.; formal analysis, E.J.M., P.K.L., V.L.T. and P.M.J.D.; investigation, V.L.T., F.N., E.J.M. and P.K.L.; resources, F.N., B.P.K. and P.M.J.D.; data curation, V.L.T., E.J.M. and P.K.L.; writing—original draft preparation, V.L.T., F.N. and E.J.M.; writing—review and editing, B.P.K., P.K.L. and P.M.J.D.; visualization, B.P.K. and P.K.L.; supervision, F.N. and P.M.J.D.; project administration, F.N., B.P.K. and P.M.J.D.; funding acquisition, F.N., B.P.K. and P.M.J.D. All authors have read and agreed to the published version of the manuscript.

**Funding:** This research received no external funding.

**Institutional Review Board Statement:** Not applicable.

**Informed Consent Statement:** Not applicable.

**Data Availability Statement:** Data are contained within the article and supplementary materials.

**Acknowledgments:** The authors acknowledge the University of Douala, Douala, Cameroon, and the University of the Mountain, Bangangté, Cameroon, for providing necessary facilities for chemical and biological tests.

**Conflicts of Interest:** The authors declare no conflicts of interest.

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
