# Peer review of "Susceptibility Patterns in Staphylococcus and Klebsiella Causing Nosocomial Infections upon Treatment with E-Anethole-Rich Essential Oil from Clausena anisata"

_ddc, doi:10.3390/ddc3010014_

Round 1
Reviewer 1 Report
Comments and Suggestions for Authors
- The title - "Susceptibility patterns in Staphylococcus and Klebsiella causing nosocomial infections upon treatment with E-anethole-rich essential oil from Clausena anisate", could be shortened, especially since the essential oil was not enriched with anethole.
- Abstract: the presentation of working methods is too broad, with details related to "Working methods".
- Complete the Introduction with more scientific data about the chemical composition of the leaves and the essential oil of Clausena anisata
- Fill in the Introduction as much recent scientific data about the therapeutic actions of the Clausena anisata species and the essential oil.
- Replace in the text with C. anisata.
- Table 2 would not be necessary.
- 2.2.1. Extraction of the essential oil - specify that the material is fresh.
- How many ml of essential oil did you get? Why didn't you use oil and dry leaves for insulation? The yield and perhaps the chemical composition of the volatile oil would have been different and also the intensity of the antimicrobial activity.
- For a better chromatographic evaluation, the GC-MS method would have been necessary.
- For microbiological testing, how was the essential oil used? What were the dilutions? What concentrations of oil were used?
- Conclusions - present only the results and less discussion already presented about nosocomial infections.
Author Response
Manuscript ID: ddc-2854083
Title: Susceptibility patterns in Staphylococcus and Klebsiella causing nosocomial infections upon treatment with E-anethole-rich essential oil from Clausena anisata.
Response to the reviewers’ comments point by point:
Reviewer 1
The title - "Susceptibility patterns in Staphylococcus and Klebsiella causing nosocomial infections upon treatment with E-anethole-rich essential oil from Clausena anisate", could be shortened, especially since the essential oil was not enriched with anethole.
Reply: After analysis of the phytochemical composition of the essential oil, which was prepared from Clausena anisata leaves, it was found that be oil was dominated by the monoterpene E-anethole (70.77%). That is why we have mentioned this essential oil as the E-anethole-rich oil across the manuscript.
- Abstract: the presentation of working methods is too broad, with details related to "Working methods".
Reply: The working methods have now been detailed in the abstract. The changes have been colored red with yellow stripes.
- Complete the Introduction with more scientific data about the chemical composition of the leaves and the essential oil of Clausena anisata
Reply: We agree with the reviewer’s apprehension. Scientific data about the chemical composition of the leaves and the essential oil of Clausena anisata have now been incorporated in the introduction section. The changes have been colored red with yellow stripes.
- Fill in the Introduction as much recent scientific data about the therapeutic actions of the Clausena anisata species and the essential oil.
Reply: As suggested, scientific data about the therapeutic actions of the Clausena anisata species and the essential oil have now been incorporated. The changes have been colored red with yellow stripes.
- Replace in the text with C. anisata.
Reply: We agree with the reviewer’s inquisitiveness. We have now incorporated C. anisata in the main text as suggested.
- Table 2 would not be necessary.
Reply: Table 2 has now been omitted.
- 2.2.1. Extraction of the essential oil - specify that the material is fresh.
Reply: Thank you. It has now been specified in the main text that the material is fresh.
- How many ml of essential oil did you get? Why didn't you use oil and dry leaves for insulation? The yield and perhaps the chemical composition of the volatile oil would have been different and also the intensity of the antimicrobial activity.
Reply: Thank you for the suggestions. As already discussed in the main text, from 4600 g of plant material, 23.46 g of essential oil was obtained (Yield of extraction: 0.51%). Future studies will encompass a detailed investigation using oil and dry leaves for insulation so as to improve the antimicrobial activity.
- For a better chromatographic evaluation, the GC-MS method would have been necessary.
Reply: Thank you for the suggestions.
- For microbiological testing, how was the essential oil used? What were the dilutions? What concentrations of oil were used?
Reply: Thank you. As already discussed in the material and methods section, the antimicrobial parameter (minimum inhibitory concentration (MIC) of the samples were evaluated using the broth microdilution method as previously described in protocol M07-A9 of the Clinical and Laboratory Standard Institute (CLSI, 2012). Thus, four microliters (4 µL) of sample were added into the first wells (already containing 196 µL of Mueller Hinton Broth (MHB)) of a 96 well plate and serial two-fold dilutions were made to achieve final concentrations ranging from 1000 to 0.048 µg/mL for the test samples and from 3.906 to 0.0038 µg/mL for ciprofloxacin (the positive control).
- Conclusions - present only the results and less discussion already presented about nosocomial infections.
Reply: We agree with the reviewer’s suggestion. The changes have been amended as recommended.
We request for re-evaluation and kind consideration.

Reviewer 2 Report
Comments and Suggestions for Authors
Dear authors, I reviewed in detail the paper entitled "Susceptibility patterns in Staphylococcus and Klebsiellacausing nosocomial infections upon treatment with E-anethole-rich essential oil from Clausena anisata".
These are my comments and suggestions.
Authors use the abbreviation for essential oil (EO) only in the abstract and no longer in the manuscript. Correct it.
In the introduction, authors should write something about essential oil extraction techniques, and what are the advantages and disadvantages of classic extraction techniques compared to modern extraction techniques.
Line 133: Remove Table 1. It is unnecessary to list the same table twice.
Line 141-143: Write the manufacturers, city and country of origin of all reagents you used.
Line 161: (Section 2.2.2) Rename to GC-MS analysis of essential oil of C. anisata. it is more suitable for the analysis of the compounds present in the essential oil. Phytochemical analysis is a broader term that includes the analysis of both volatile and non-volatile components.
Section 2.2.2 should be better structured, shorter and more precisely written.
it is only necessary to write the full name in the manuscript the first time, after that you can use the mentioned abbreviations. (EO, MIC, MBC, GC-MS, C. anisata…) Check it out in manuscript.
In section 2.3. it is twice written: data were expressed as mean ± standard deviation.
Section 3.1.1. There is no need to repeat the same results in the text and the Table 2. Decide on one.
Consider moving Figure 2 to supplementary material.
Line 301: Remove MHB from the legend of Table 4.
It is not usual to have references in the conclusion. Maybe it's better to move the reference to the discussion section.
Make sure all references are well written.
The manuscript needs to be improved, make the mentioned modifications. English is understandable. I suggest acceptance after accepting all comments. I suggest acceptance after major revision.
Kind regards

Author Response
Manuscript ID: ddc-2854083
Title: Susceptibility patterns in Staphylococcus and Klebsiella causing nosocomial infections upon treatment with E-anethole-rich essential oil from Clausena anisata.
Reviewer 2
Dear authors, I reviewed in detail the paper entitled "Susceptibility patterns in Staphylococcus and Klebsiella causing nosocomial infections upon treatment with E-anethole-rich essential oil from Clausena anisata".
Dear Reviewer,
We are very thankful for your efforts and the valuable comments that you have suggested for our manuscript.
We have now revised the manuscript in accordance with your comments. The point by point response to the comments are as follows:
These are my comments and suggestions.
Authors use the abbreviation for essential oil (EO) only in the abstract and no longer in the manuscript. Correct it.
Reply: We agree with the reviewer’s apprehension. The change has been amended as suggested.
In the introduction, authors should write something about essential oil extraction techniques, and what are the advantages and disadvantages of classic extraction techniques compared to modern extraction techniques.
Reply: As suggested, we have now incorporated writings about extraction techniques of essential oil in the introduction section.
Line 133: Remove Table 1. It is unnecessary to list the same table twice.
Reply: The changes have been amended as suggested. Table1 has now been omitted.
Line 141-143: Write the manufacturers, city and country of origin of all reagents you used.
Reply: Information (manufacturers, city and country of origin) related to all reagents used have now been incorporated at the appropriate places of the main text.
Line 161: (Section 2.2.2) Rename to GC-MS analysis of essential oil of C. anisata. it is more suitable for the analysis of the compounds present in the essential oil. Phytochemical analysis is a broader term that includes the analysis of both volatile and non-volatile components.
Reply: We agree with the reviewer’s suggestion. We have now renamed section 2.2.2. as recommended.
Section 2.2.2 should be better structured, shorter and more precisely written.
Reply: As suggested, Section 2.2.2 has now been rewritten. The changes have been colored red with yellow stripes.
it is only necessary to write the full name in the manuscript the first time, after that you can use the mentioned abbreviations. (EO, MIC, MBC, GC-MS, C. anisata…) Check it out in manuscript.
Reply: The changes has been amended as suggested.
In section 2.3. it is twice written: data were expressed as mean ± standard deviation.
Reply: We agree with the reviewer’s inquisitiveness. We have now corrected the mistake.
Section 3.1.1. There is no need to repeat the same results in the text and the Table 2. Decide on one.
Reply: We agree with the reviewer’s apprehension. Table 2 has now been omitted.
Consider moving Figure 2 to supplementary material.
Reply: Figure 2 has now been incorporated as a supplementary material.
Line 301: Remove MHB from the legend of Table 4.
Reply: The change has been amended as suggested.
It is not usual to have references in the conclusion. Maybe it's better to move the reference to the discussion section.
Reply: The changes have been amended as recommended. The reference used in the conclusion section has now been omitted.
Make sure all references are well written.
Reply: We have checked to make sure that all references are well written. However, we are ready to accept any other changes if required.
The manuscript needs to be improved, make the mentioned modifications. English is understandable. I suggest acceptance after accepting all comments. I suggest acceptance after major revision.
Kind regards
Reply: We have now implemented all the corrections as indicated and we thank you very much for the efforts that you have made to improve our manuscript.
We request you for re-valuation and kind consideration.
With best regards,

Round 2
Reviewer 1 Report
Comments and Suggestions for Authors
The authors completed and revised the manuscript. Thank you!
Reviewer 2 Report
Comments and Suggestions for Authors
Dear authors, I have reviewed in detail revised version of the manuscript entitled „Susceptibility patterns in Staphylococcus and Klebsiella causing nosocomial infections upon treatment with E-anethole-rich essential oil from Clausena anisata“ I agree with all the changes made in manuscript. The manuscript can be accept in present form.
Kind regards
